# Transferable Graph Optimizers for ML Compilers

Yanqi Zhou[1], Sudip Roy[1], Amirali Abdolrashidi[2], Daniel Wong[3], Peter Ma[1], Qiumin Xu[1],
Hanxiao Liu[1], Mangpo Phitchaya Phothilimtha[1], Shen Wang[1], Anna Goldie[1],
Azalia Mirhoseini[1], and James Laudon[1]

[1]Google, Mountain View, CA, USA
{yanqiz, sudipr, pcma, qiuminxu, hanxiaol, mangpo, shenwang, agoldie,
azalia, jlaudon}@google.com
[2]UC Riverside, Riverside, CA, USA
abdolrashidi@gmail.com
[3]Carnegie Mellon University, Pittsburgh, PA, USA
wonglkd@gmail.com

## Abstract

Most compilers for machine learning (ML) frameworks need to solve many correlated optimization problems to generate efficient machine code. Current ML compilers rely on heuristics based algorithms to solve these optimization problems one at a time. However, this approach is not only hard to maintain but often leads to sub-optimal solutions especially for newer model architectures. Existing learning based approaches in the literature are sample inefficient, tackle a single optimization problem, and do not generalize to unseen graphs making them infeasible to be deployed in practice. To address these limitations, we propose an end-to-end, transferable deep reinforcement learning method for computational graph optimization (GO), based on a scalable sequential attention mechanism over an inductive graph neural network. GO generates decisions on the entire graph rather than on each individual node autoregressively, drastically speeding up the search compared to prior methods. Moreover, we propose recurrent attention layers to jointly optimize dependent graph optimization tasks and demonstrate 33%-60% speedup on three graph optimization tasks compared to TensorFlow default optimization. On a diverse set of representative graphs consisting of up to 80,000 nodes, including Inception-v3, Transformer-XL, and WaveNet, GO achieves on average 21% improvement over human experts and 18% improvement over the prior state of the art with $15\times$ faster convergence, on a device placement task evaluated in real systems.

## 1 Introduction

Increasingly, many applications are driven by large and complex neural network models [12, 28, 15, 18, 24]. The high computation requirements of training such models requires efficient use of ML accelerator(like GPUs and TPUs). However, the effective use of such accelerators is largely determined by device-specific optimization by compilers, like TensorFlow XLA, Glow, MLIR, and AutoTVM [10, 26, 17, 4], which map the high-level computational graph to operations executable on the device.

In mapping a computational graph to machine code that executes on a collection of devices, ML compilers need to solve many optimization problems including graph rewriting, assignment of operations on devices, operation fusion, layout and tiling of tensors, and scheduling. ML compilers usually apply heuristics to solve these problems individually, which suffers from two key limitations.

First, these heuristics often lead to sub-optimal configurations especially for previously unseen model architectures. Second, by solving these problems in isolation, the compiler misses out on opportunities for joint optimizations across tasks. To overcome these limitations of current approaches, ML practitioners often rely on their domain knowledge and use manual hints (for example, explicit device assignments) to guide the compiler's decisions.

Prior RL-based approaches [21, 19, 9] outperform both human experts as well as heuristic algorithms. However, they often require substantial computational resources to train and do not generalize well to new graphs. Furthermore, most prior learning based approaches are targeted towards solving a single optimization problem in the stack without any knowledge sharing across tasks. Many of the graph optimization problems in the compiler stack are inherently coupled. For example, a seemingly well optimized graph partitioning and device placement can lead to poor run time due to bad scheduling decisions that induces a near-sequential execution. For making learned solutions practically viable and competitive, we need designs that are not only resource efficient and fast but also are able to jointly solve tightly coupled optimization problems in the stack.

In this paper, we propose an end-to-end deep RL method (GO) for ML compiler graph optimizations where the learned policy is generalizable to new graphs and transferable across multiple tasks. Specifically, GO consists of an inductive graph-embedding network that encodes operation features and dependencies in a trainable graph representation, followed by a policy network of segmented recurrent attention layers. The policy network transforms the graph representations into an optimization decision with soft attention. These two networks can be jointly trained in an end-to-end fashion using a supervised reward. To generalize to arbitrary and held-out graphs, GO is trained jointly over a set of computation graphs (instead of one at a time) and then fine-tuned on new graphs. By transferring the learned graph embeddings and optimization policies, GO converges faster using less resources. We also use super-positioning, a feature conditioning mechanism based on the input graph embeddings, to effectively orchestrate the optimization dynamics of a batch containing graphs with drastically different sizes. To jointly optimize multiple dependent graph optimization tasks, we propose a novel recurrent attention, without introducing additional parameters or training overhead.

## 2   Related Work

**Model Level Parallelism** Mesh-TensorFlow is a language that provides a general class of distributed tensor computation, where users can specify any tensor-dimension to be split across any dimension of a multi-dimensional mesh of processors. FlexFlow [14] introduces SOAP, a more comprehensive search space of parallelization strategies for DNNs which allows parallelization of a DNN in the Sample, Operator, Attribute, and Parameter dimensions. GPipe [13] proposed pipeline parallelism by partitioning a model across multiple accelerators and automatically splitting a mini-batch of training examples into smaller micro-batches. PipeDream [22] introduced pipeline parallelism with asynchronous training, allowing gradient updates of multiple mini-batches to happen in parallel. In addition to the model parallelism primitives, GO provides a learning-based optimization that can generalize across different graphs and transfer to new tasks.

**RL for Graph Optimization** Reinforcement learning has been used for device placement [21, 9, 19] and has demonstrated run time reduction over human-crafted placements and conventional heuristics. Hierarchical Device Placement (HDP) [19], Spotlight [9], and Placeto [1] progressively generate decisions on a per-node basis, so they have difficulty capturing long-distance dependencies over large graphs and are slow in training. Placeto [1] represents the first attempt to generalize device placement using a graph embedding network. However, like HDP, Placeto relies on hierarchical grouping and only generates placement for one node at each time step. NeuRewriter [5] is a generic rewriting system that uses RL to solve combinatorial optimization problems. It performs a series of local rewrites until reaching a final solution. In contrast, our approach generates decisions for the entire graph at once, significantly reducing the policy training time. We also tackle a wider set of graph optimization problems.

**ML Compiler Optimization** Machine learning has been applied to optimize the execution time of tensor computation graphs [23, 34, 2]. Out of all these works, only REGAL[23] leverages the learned policy's ability to transfer knowledge to new graphs. However, we demonstrate generalization on substantially larger and varied set of graphs, Unlike REGAL, we evaluate GO with performance measurements on real systems in addition to using a performance model. TASO [36] automatically

generates graph substitutions for DNN computational graphs. However, TASO is not learning-based and does not generalize to unseen graphs. FlexFlow [14] introduces SOAP, a more comprehensive search space of parallelization strategies for DNNs which allows parallelization of a DNN in the Sample, Operator, Attribute, and Parameter dimensions.

## 3 Tasks and Problem Formulation

ML computations are usually expressed as computation graphs, $G(V, E)$, where nodes $V$ represent computations and edges $E$ represent data flows. We consider three related computational graph optimization problems from the ML compiler optimization stack. We first explain how each of these problems can be formulated as learning a policy for classification of nodes in the graph. We then discuss the specific training objectives used in GO.

**Device Placement:** Given a computational graph, the goal of device placement is to learn a policy $\pi : \mathcal{G} \mapsto \mathcal{D}$ that assigns a device $D \in \mathcal{D}$ for all nodes in the given graph $G \in \mathcal{G}$, to maximize a reward $r_{G,D}$ defined based on the run time.

**Operation Scheduling:** An op in a dataflow graph is *ready* to run when its incoming tensors are present in the device memory. A frequently used scheduling strategy is to maintain a ready queue of operations for each device and schedule operations in first-in-first-out order. However, such schedules can be sub-optimal especially when the operations are distributed across a set of devices. As demonstrated through detailed profiles in Supp. Mat. B.1, the sub-optimal schedules typically exhibit underutilized devices since ops for these devices are blocked on producer ops in ready queues of other devices. We propose a priority-based scheduler where the scheduler maintains a *priority* queue of ready operations for each device and schedules the highest priority operation in the queue first. Similar to device placement, operation scheduling can be formulated as the problem of learning a policy $\pi : \mathcal{G} \mapsto \mathcal{P}$ that assigns a scheduling priority $P \in \mathcal{P}$ for all ops in the graph to maximize a reward $r_{G,P}$ defined based on run time.

**Operation Fusion:** Op fusion is the process of merging multiple ops into a single op. Figure 1 showcases an example of op fusion. Op $K1$ is producing an output which is consumed by op $K2$. If these two ops are fused, the intermediate data produced by $K1$ is immediately used for $K2$ when $K1$ finishes on the device, without the need to perform read and write transactions with the global memory, thereby reducing the communication overhead. Figure 2 shows an example of how the decision of which ops to fuse can change application performance. In this case, we have an element-wise multiplication (Mul), a reduction, and a sigmoid function connected to each other as shown. Should the algorithm choose Reduce and Sigmoid for fusion (left), the performance will not improve much, since the amount of intermediate values transferred to/from the memory will not change significantly. On the other hand, if the fusion algorithm selects Mul and Reduce (right), the intermediate tensor after the multiplication will stay in the faster scratchpad memory for the Reduce op. Therefore, the amount of transferred data from/to global memory has decreased dramatically. Simple strategies like considering ops in topological order can make inefficient fusion decisions and lead to suboptimal performance. We can reformulate op fusion as a priority-based ordering problem: learning a policy $\pi : \mathcal{G} \mapsto \mathcal{F}$ that assigns a fusion priority $F \in \mathcal{F}$ for all ops in the graph to maximize a reward $r_{G,F}$ defined based on run time. $r_{G,F}$. Each node is associated with a fusion priority score $F \in \mathcal{F}$, where the size of $\mathcal{F}$ is a hyperparameter. The scores determine the order in which nodes are fused.

**Generalization across graphs:** In contrast to prior works that focus on a single graph only, the training objective for GO is to simultaneously reduce the expected run time of the optimization over

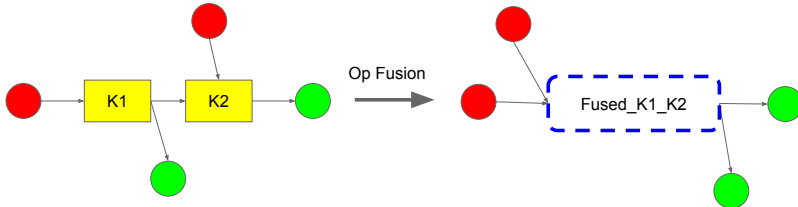

Figure 1: An op fusion example.

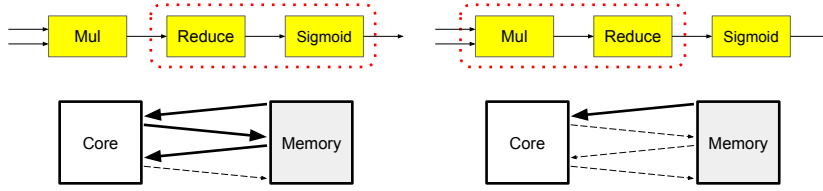

Figure 2: Inefficient fusion with unnecessary memory traffic (left), and better fusion (right).

multiple dataflow graphs: $J(\theta) = \mathbb{E}_{G \sim \mathcal{G}, T \sim \pi_\theta(G)}[r_{G,T}]$, where $T \in \{D, P, F\}$ denotes the task, $\theta$ denotes the parameters of the RL policy, and $\mathcal{G}$ denotes the empirical distribution of dataflow graphs. Figure 3 demonstrates the overall network design for a single task.

**Joint optimization across tasks:** Our method can be extended to handle multiple tasks jointly. The multi-task objective is defined as: $J(\theta) = \mathbb{E}_{G \sim \mathcal{G}, D \sim \pi_{\theta_D}(G), P \sim \pi_{\theta_P}(G), F \sim \pi_{\theta_F}(G)}[r_{G,D,P,F}]$. Parameters across the three tasks can be partially shared: $\theta_T = (\phi, \psi_T)$ where $T \in \{D, P, F\}$, $\phi$ denotes the shared parameters and $\phi_T$ denotes the task-specific parameters. The shared policies can be parameterized in the form of multiple recurrent attention layers (Section 4.4).

# 4 Network Architecture

In this section, we first present the architecture of a policy network that can be adapted to solve each of the fore-mentioned optimization problems one at a time. In Section 4.4, we further present the augmentation with recurrent attention layers to learn multiple tasks jointly.

Figure 3 shows an overview of the proposed end-to-end graph policy network. Our proposed policy network $\pi_\theta$ consists a graph embedding network that learns the graphical representation $h_G$ of any computational graph, and a scalable decision network that learns a optimization strategy $p(\mathbf{y}|G)$ over the given graph representation. The two components are jointly trained in an end-to-end fashion.

Let $y_i$ be the action for the $i$-th node. Ideally, we would like to compute the action distribution of the current node based on the actions of all previous nodes in an auto-regressive manner:

$$p(\mathbf{y}|G) = \prod_{i=1..N} p\left(y_i | h_G, y_{i-1}, y_{i-2}, ...\right) \tag{1}$$

However, the above is infeasible in practice because $N$ can be as large as 10K, and computing the $y_i$'s sequentially can be extremely expensive. To address this issue, we propose an iterative but

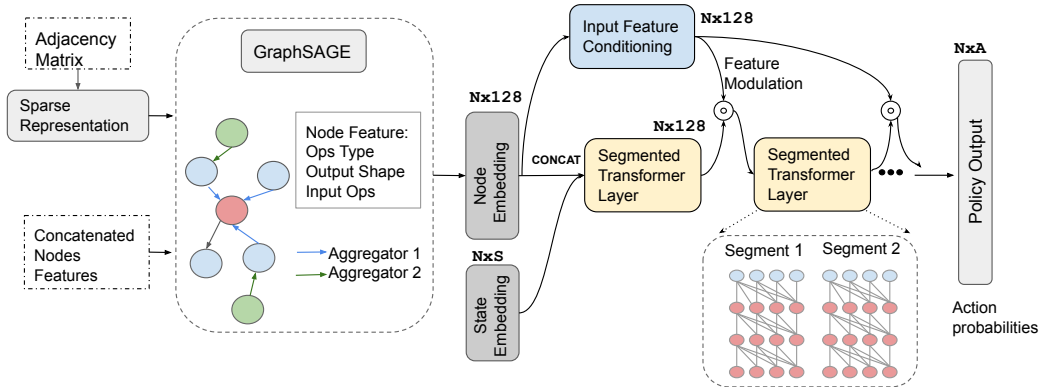

Figure 3: Overview of GO: An end-to-end graph policy network that combines graph embedding and sequential attention. $N$: Number of Nodes, $a$: Size of the action space (number of devices, number of priority levels, etc.). Node features are sorted in topological order.

non-autoregressive process as an approximation:

$$p(\mathbf{y}^{(t)}|G) = \prod_{i=1..N} p\left(y_i^{(t)}|h_G, \mathbf{y}^{(t-1)}\right) \tag{2}$$

Although the $N$ sampling procedures are now carried out in parallel within each iteration $t$, decisions over the $N$ nodes are allowed to mutually influence each other because the process above will be repeated for $T$ times ($T \ll N$). Note the distribution of $\mathbf{y}^{(t)}$ is informed about $\mathbf{y}^{(t-1)}$, the actions made over all the nodes in the previous iteration.

In terms of network implementation, the temporal dependencies between actions in multiple RL steps are carried out by state embedding such that the state of the previous step is used as an input to the network in the next step. In Eq. 1, the actions are produced by the final layer in Figure 3. Figure 3 is an illustration of a single task, while Figure 4 demonstrates the extension of the same framework for multiple tasks. We add a few more Transformer layers with residual connections as task heads.

The architecture is designed to be invariant over the underlying graph topology, enabling us to apply the same learned policy to a wide set of input graphs. GO optimizes the objective described in Section 3 using Proximal Policy Optimization (PPO) [27] for improved sample efficiency.

## 4.1 Graph Embedding Network

We leverage graph neural networks (GNNs) [11, 32, 33] to capture the topological information encoded in the dataflow graph. GraphSAGE [11] is an inductive network that leverages node attribute information to generalize to previously unseen data. While our proposed framework is generic, we adopt the feature aggregation scheme proposed in GraphSAGE and build a general, end-to-end graph optimization method for a wide set of dataflow graphs.

In GO, nodes and edges in the dataflow graph are represented as the concatenation of their meta features (e.g., operation type, output shape) and are further encoded by the graph embedding network into a trainable representation. The graph embedding process consists of multiple iterations, and the computation procedure for the $l$-th iteration can be outlined as follows: First, each node $v \in V$ aggregates the feature representations of its neighbors, $\{h_u^{(l)}, \forall u \in \mathcal{N}(v)\}$, into a single vector $h_{\mathcal{N}(v)}^{(l)}$. This aggregation outcome is a function of all previously generated representations, including the initial representations defined based on the input node features. In this work, we use the following aggregation function with max pooling:

$$h_{\mathcal{N}(v)}^{(l)} = \max(\sigma(W^{(l)}h_u^{(l)} + b^{(l)}), \forall u \in \mathcal{N}(v)) \tag{3}$$

where $(W^{(l)}, b^{(l)})$ define an affine transform and $\sigma$ stands for the sigmoid activation function. We then concatenate the node's current representation, $h_v^{(l)}$, with the aggregated neighborhood vector, $h_{\mathcal{N}(v)}^{(l)}$, and feed this concatenated vector through a fully connected layer $f^{(l+1)}$

$$h_v^{(l+1)} = f^{(l+1)}(\text{concat}(h_v^{(l)}, h_{\mathcal{N}(v)}^{(l)})) \tag{4}$$

Different from GraphSAGE, parameters in our graph embedding network are trained jointly with a decision network via stochastic gradient descent with PPO, in a *supervised* fashion. Enhanced with a global attention network discussed in Section 4.2, the optimal GS layers is smaller.

## 4.2 Scalable Decision Network

In graph optimization tasks, the optimal actions for a node are often influenced by actions for other nodes in the graph. While the graph neural network works as a feature aggregation network, it lacks the ability of tracking global node dependencies in a scalable fashion. Intuitively, an attention network can learn this dependency and the relative importance of dependencies across an entire graph.

In addition to tracking dependencies in a scalable fashion, handling large computational graphs in the compilation stacks is another practical concern when designing the network. The decision network should be able to handle computational graphs from realistic workloads consisting over 10,000 nodes. As compared in Table 1, LSTM-based models proposed for language tasks usually target a shorter sequence length, incurring vanishing (and exploding) gradients or substantially longer

| Architectures | Handle over 10k nodes | Capture topological info | Global node dependencies | Generalizable | Fast |
| --- | --- | --- | --- | --- | --- |
| No GraphSAGE | - | - | - | No | - |
| GS+MLP | Yes | Yes | No | Yes | Yes |
| GS+LSTM | No | Yes | No | Yes | No |
| GS+VanillaTransformer | No | Yes | Yes | Yes | Yes |
| Graph Attention | No | Yes | No | Yes | Yes |
| **GO** | **Yes** | **Yes** | **Yes** | **Yes** | **Yes** |

Table 1: Network Comparison. GO is composed of a GraphSAGE with a segmented Transformer Network that can satisfy all the listed requirements.

training time. Although hierarchical grouping has been used [19, 1] to address longer sequences in a LSTM-based network, the proposed grouper network comes with limited flexibility and generality. The non-differentiable grouping procedure prevents training the networks end-to-end.

The above requirements and considerations led us to propose a Transformer-based attentive network to generate the optimization decision in an end-to-end fashion as shown in Figure 3. As the graph embedding already contains spatial (topological) information for each node, we remove the positional embedding in the original transformer to prevent the model from overfitting node identifications. To capture long-term dependencies efficiently among a large set of nodes, we adopt segment-level recurrence introduced in Transformer-XL [8, 7], where hidden states computed for the previous set of nodes are cached (with gradient flows disabled) and reused as an extended context during the training of the next segment. Besides achieving extra long context, we empirically find the segment-level recurrent attention much faster than a LSTM-based method. In our experimental evaluation, we compare both the performance and speed up of our policy network with that of the LSTM-based hierarchical policy network. More ablation study is presented in Supp. Mat. C.1.

## 4.3 Generalization with Parameter Superposition

One of the key goals of this work is to ensure the generalizability of the the policy network over a wide variety of graphs from potentially different application domains (e.g. computer vision, language, and speech). Not only do these graphs vary in the number of operations from a few thousand to a million, but they also have drastically different network architectures, in terms of computational operations, data shape, and network topology. As an example, recurrent networks have completely different operation types and connections compared to convolutional networks that are widely used in computer vision. A naïve strategy of training the shared policy network with batches of heterogeneous graphs is unlikely to perform as well as networks exclusively trained for a particular type of graph.

To overcome this limitation, we propose a feature modulation mechanism similar to *parameter superposition* [6]. The idea is to re-weight network parameters by generating a feature modulation layer based on the input graph features as shown in Figure 3, to mitigate the potentially undesirable interference among different input graphs. The feature modulation layer is dot multiplied with all intermediate feature maps in the decision network: $x^{(l+1)} = a^{(l)}(m(h_G) \odot x^{(l)})$, where $a^{(l)}$ stands for an attention layer in our policy network, $m$ stands for the feature modulation layer, and $h_G$ is the feature representation of the input graph generated by the graph-embedding network. The feature modulation layer is implemented with minimum overhead by adding an additional transformer layer to our decision network. The output of the modulation layer is defined as the last-layer features of the additional transformer.

## 4.4 Multi-task Policy

Given the generality of the policy network presented in the previous section, it is easy to now extend it jointly solve multiple dependent graph optimization tasks without introducing significant parameters or training time. We propose a recurrent attention policy network that not only applies a segment-level recurrent attention to the graph representation spatially, but also generates recurrent actions for multiple tasks through residual connections and parameter sharing across multiple recurrent attention layers. Compared to the single-task policy net, Figure 4 shows that multi-task GO only introduces one recurrent attention layer for each task added and the parameters are shared across different tasks. The recurrent attention layers with residual connections of actions enables tracking inter-task dependencies.

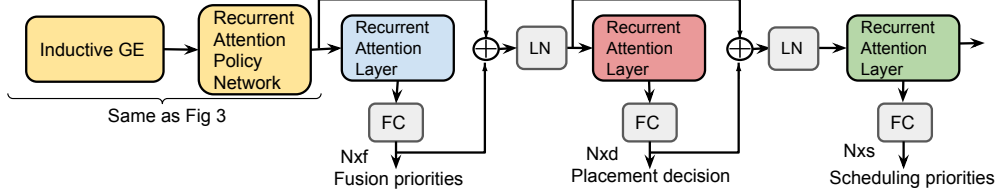

Figure 4: Multi-task policy network with additional recurrent attention layers for each task and residual connections. LN and FC stand for layer normalization and fully-connected layer, respectively.

Figure 4 shows the multi-task policy network consisting of a spatial recurrent attention that attend to the input sequence (the recurrent attention policy network) and temporal recurrent attention that attend to the task sequence (the multi-task recurrent attention layers). For the $t$-th task, the recurrent attention layer computes the its hidden state $H^t$ and action $A^t$ as follows

$$H^t = \text{LN}(\text{Concat}(A^{t-1}, H^{t-1})) \tag{5}$$

$$A^t = \text{FC}(\text{MultiHeadAttn}(H^t)) \tag{6}$$

where LN stands for layer normalization [3], MultiHeadAttn stands for a multi-head attention layer whose parameters are shared across layers. FC is a fully-connected layer that consists of a single rectified-linear activation between two affine transformations, applied position-wise, that projects the hidden state to the task action space.

## 5 Experimental Evaluation

For each of the computational graph optimization tasks, the end goal is to minimize the step time of the graph or a batch of dataflow graphs for a target system configuration (e.g. a 8-GPU cluster or a TPU cluster), by learning a policy that generates optimization decisions on a per-node basis.

The performance of an optimized graph is evaluated by the resulting step time, run time for a single iteration through the graph measured either by executing the graph on real hardware or using an accurate simulator. We use the negative square root of the normalized run time as the reward, where the run time is normalized with the best run time from a baseline. We use a value network to generate the bias term. The advantage value is computed by subtracting the return by the bias turn. During the search, we apply a large negative reward (-10) for invalid optimizations (e.g. a violation of co-location constraint, out of memory, etc.).

For better sample efficiency, we adopted a Proximal Policy Optimization (PPO) [27] algorithm. Within a loop, PPO continuously samples actions from the distribution and evaluates the step times. For a rollout of $K$, we perform a minibatch of $m$ stochastic gradient ascent steps with respective to the objective of PPO. We find a set of optimized hyper parameters and keep them fixed for all the experiments presented. The optimal found PPO hyper parameters are presented in Supp. Mat. A.1.

**Workloads:** We evaluate GO using the computational graphs of six diverse architectures from different domains. Specifically, we use LSTM-based RNN Language Model [35, 15], GNMT [29], and Transformer-XL [8] from language domain; InceptionV3 [30] and AmoebaNet [25] from computer vision; and finally WaveNet [31] from the speech domain. To create a larger set of workloads we vary architectural parameters like the number of layers for each of these workloads. All our workloads are implemented in TensorFlow. Further details about the graphs is in Supp. Mat. A.2.

**Run Time Measurement:** For placement task, where TensorFlow provides an API for device assignment, our experiments are evaluated on actual hardware with configuration of one Intel Broadwell CPU and up to eight Nvidia P100 GPUs. For fusion and scheduling tasks, where an API for setting nodes' priorities is not available in TensorFlow, we instead use an analytical performance model based on roofline estimates (details in Supp. Mat. A.3) for evaluation. For all results presented, we use an average result over six runs for both baseline models and our method.

**Baselines:** We choose four different baselines against which we compare the performance of GO along various metrics. They are the default heuristics-based optimizations used in TensorFlow(GPU), a human-expert solution, solutions found using non-learning search techniques like simulated annealing, and finally solutions found using a learning based strategy like HDP [19].

| Model (#devices) | GO-one (s) | HP (s) | METIS (s) | HDP (s) | Run time speed up over HP / HDP | Search speed up over HDP |
|---|---|---|---|---|---|---|
| 2-layer RNNLM (2) | 0.173 | 0.192 | 0.355 | 0.191 | 9.9% / 9.4% | 2.95x |
| 4-layer RNNLM (4) | 0.210 | 0.239 | 0.503 | 0.251 | 13.8% / 16.3% | 1.76x |
| 8-layer RNNLM (8) | 0.320 | 0.332 | OOM | 0.764 | 3.8% / 58.1% | 27.8x |
| 2-layer GNMT (2) | 0.301 | 0.384 | 0.344 | 0.327 | 27.6% / 14.3% | 30x |
| 4-layer GNMT (4) | 0.350 | 0.469 | 0.466 | 0.432 | 34% / 23.4% | 58.8x |
| 8-layer GNMT (8) | 0.440 | 0.562 | OOM | 0.693 | 21.7% / 36.5% | 7.35x |
| 2-layer Transformer-XL (2) | 0.223 | 0.268 | 0.37 | 0.262 | 20.1% / 17.4% | 40x |
| 4-layer Transformer-XL (4) | 0.230 | 0.27 | OOM | 0.259 | 17.4% / 12.6% | 26.7x |
| 8-layer Transformer-XL (8) | 0.350 | 0.46 | OOM | 0.425 | 23.9% / 16.7% | 16.7x |
| Inception (2) b32 | 0.229 | 0.312 | OOM | 0.301 | 26.6% / 23.9% | 13.5x |
| Inception (2) b64 | 0.423 | 0.731 | OOM | 0.498 | 42.1% / 29.3% | 21.0x |
| AmoebaNet (4) | 0.394 | 0.44 | 0.426 | 0.418 | 26.1% / 6.1% | 58.8x |
| 2-stack 18-layer WaveNet (2) | 0.317 | 0.376 | OOM | 0.354 | 18.6% / 11.7% | 6.67x |
| 4-stack 36-layer WaveNet (4) | 0.659 | 0.988 | OOM | 0.721 | 50% / 9.4% | 20x |
| GEOMEAN | - | - | - | - | **20.5% / 18.2%** | **15x** |

Table 2: Run time comparison between GO-one, human expert, TensorFlow METIS, and hierarchical device placement (HDP) on six graphs (RNNLM, GNMT, Transformer-XL, Inception, AmoebaNet, and WaveNet). Search speed up is the policy network training time speed up compared to HDP (reported values are averages of six runs).

| Speedup | TF default | SA | GO-one | Speedup | TF default | SA | GO-one |
|---|---|---|---|---|---|---|---|
| NMT (2GPU) | 2.82 | 3 | **3.19 (+0.37)** | RNNLM (8GPU) | -2.39 | -2.38 | **-2.27 (+0.11)** |
| NMT (4GPU) | -0.89 | 5.34 | **12.03 (+12.92)** | TRF-XL (2GPU) | 24.27 | 25.1 | **28.51 (+4.24)** |
| NMT (8GPU) | 10.47 | 10.47 | **12.65 (+2.18)** | TRF-XL (4GPU) | 17.05 | 19.32 | **19.99 (+2.94)** |
| RNNLM (4GPU) | 1.04 | 1.06 | **1.23 (+0.19)** | TRF-XL (8GPU) | 21.66 | 26.25 | **31.48 (+9.82)** |

Table 3: Speedup of each fusion policy normalized to the no-fusion case (reported in %). The number in the parentheses is the improvement of our work over the default fusion.

## 5.1 Single Task Performance

**Device Placement for Individual Graphs:** To demonstrate that GO's policy network architecture is better suited for graph optimization problems, we first train the model separately on each of our workloads. We name this approach **GO-one**. Since TensorFlow provides an API for assigning operation placement, all reported measurements for placement task are on real hardware. As shown in Table 2, GO-one consistently outperforms human expert placement (HP), TensorFlow METIS [16] placement, and Hierarchical Device Placement (HDP). GO is designed in a way to scale up to extremely large graphs, consisting of over 80,000 nodes (8-layer GNMT). Therefore unlike any of the prior works including HDP [19], REGAL [23], and Placeto [1], we can demonstrate super human performance on large graphs such as 8-layer GNMT (21.7%/36.5% better than HP/HDP) and 8-layer RNNLM (3.8%/58.1% better than HP/HDP). Overall, GO-one achieves on average 20.5% and 18.2% run time reduction across the evaluated 14 graphs, compared to HP and HDP respectively. Importantly, with the efficient end-to-end single-shot placement, GO-one has a 15x speedup in convergence time of the placement network over HDP.

**Op Fusion for Individual Graphs:** We apply the same GO network to a operation fusion task for a subset of our workload graphs (namely, Transformer-XL, GNMT, and RNNLM). We estimate the run time with the the performance model described in Supp. Mat. A.3 for V100 GPUs for the following cases: no-fusion, TensorFlow GPU default fusion, priority fusion via Simulated Annealing (SA), and GO-one. Table 3 showcases the resulting speedups normalized with respect to the no-fusion run time. Compared to TensorFlow default fusion, GO improves the training step time on average by 5.7%. Compared with SA, which takes 20x[1] longer to search, GO outperforms by an average of 4.6%.

**Generalization Across Graphs:** GO enables the training of multiple heterogeneous graphs in a single batch, sharing parameters in the networks. Inspired by the pre-training and fine-tuning method, we pretrain GO over all but one workloads. We randomly sample from this set of input graphs to construct a batch. We train GO for 1000 steps for each batch before switching to the next batch. We then fine-tune the pre-trained model on the hold-out graphs (i.e., graphs from the sixth workload not

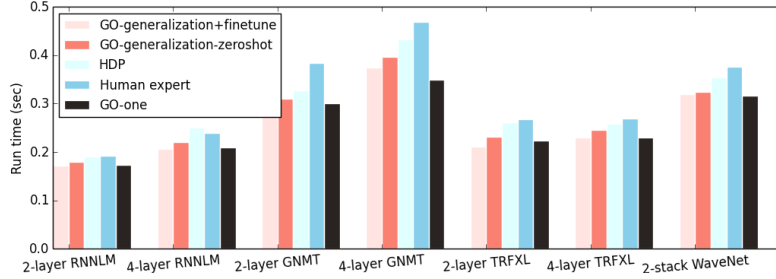

Figure 5: Generalization across heterogeneous workload graphs. The figure shows a comparison of two different generalization strategies for GO when trained with graphs from 5 of the 6 workloads, and evaluated on the held-out workload (x-axis).

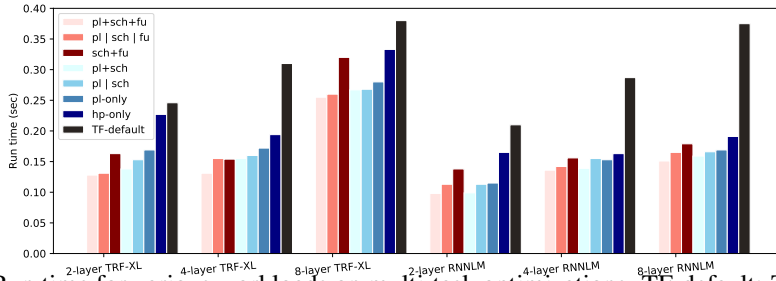

Figure 6: Run time for various workloads on multi-task optimizations. TF-default: TF GPU default placement, fusion, and scheduling. hp-only: human placement only with default scheduling and fusion. pl-only: GO placement only with default scheduling and fusion. pl|sch: GO optimizes placement and scheduling individually with default fusion. pl+sch: multi-task GO co-optimizes placement and scheduling with default fusion. sch+fu: multi-task GO co-optimizes scheduling and fusion with human placement. pl|sch|fu: GO optimizes placement, scheduling, and fusion separately. pl+sch+fu: multi-task GO co-optimizes placement, scheduling, and fusion.

included in the training set) for fewer than 50 steps, which takes less than one minute. We name this **GO-generalization+finetune**. Figure 5 shows that GO fine-tuning for hold-out graphs outperforms human expert placement and HDP consistently on all datasets, and on average matches GO-one. We test on unseen graphs from different workloads by pre-training the model on different workloads, excluding the entire workload of the unseen graph. For example, for 2-layer RNNLM, all RNNLM models are excluded from the pre-training dataset. RNNLM, 2-layer TRFXL, 2-layer GNMT even outperfrm GO-one by 1-4%. We also run inference directly on the pre-trained model for the target hold-out graphs, and name this **GO-generalization-zeroshot**. We find that GO-generalization-zeroshot only marginally hurts performance compared to GO-generalization+finetune, while being slightly better than human placement and HDP. This indicates that both graph embedding and the learned policies transfer and generalize to the unseen data.

## 5.2 Multi-task Performance

As shown in Figure 6, co-optimizing placement, scheduling, and fusion (pl+sch+fu) provides $33\% - 60\%$ speedup compared to TensorFlow default placement, scheduling, and fusion. Comparing to optimizing each tasks individually, multi-task GO (pl+sch+fu) outperforms single-task GO (pl|sch|fu) — optimizing all tasks, one at a time — by an average of 7.8%. Furthermore, for all workloads, co-optimizing all three tasks offers faster run time than optimizing any two of them and using the default policy for the third.

## 6 Conclusion

In this paper, we present a generalized deep RL method for computation graph optimizations that generalizes to arbitrary and held out graphs. We propose recurrent attention layers to jointly optimize dependent graph optimization tasks and demonstrate 33%-60% speedup over TensorFlow's default strategy on three tasks.

# 7 Broader Impact

The increasing complexity and diversity of hardware accelerators has made the development of robust and adaptable ML frameworks onerous and time-consuming, often requiring multiple years of effort from hundreds of engineers. In this paper, we demonstrated that many of the optimization problems in such frameworks can be solved efficiently and optimally using a carefully designed learned approach. This has two significant benefits over a heuristic based hand-tuned approach. First, it can potentially save years worth of engineering effort needed to design and maintain the set of heuristics with each new generation of hardware. And second, the improved solutions found using a learned approach can have a multiplicative effect by improving hardware utilization and computational efficiency for all workloads. This increased efficiency may eventually lead to a reduction in the overall carbon footprint for many applications.

We also want to highlight a broader effort in the community to use machine learning in the hardware design process[20]. The techniques presented in this paper can be instrumental in evaluating the behavior of benchmark workloads on new and unseen hardware architectures without requiring significant redesign of compilers. Therefore, we believe that the ideas introduced in this paper are an important step that can positively impact the larger ML for Systems research directions.

One of the potential downside of the work is the loss of explainability for the choices made by the learned model. An advantage of heuristic based approaches deployed in current systems is the ability to explain the choices made by the algorithm based on the heuristics used. Often it is feasible to "fix" a poor decision by designing a customized heuristic. The current learned approaches to optimization problems including the ones presented in this paper do not lend themselves to explainable results. However, a principled approach to integrating domain specific knowledge of compiler developers with the learning based approach with the goal of improving explainability is an interesting direction for future research.

## Footnotes

[1]SA takes around 24 hours to converge in our evaluated tasks. For a smaller graph, GO takes less than an hour to converge. For a large graph over 10k nodes, GO can take 1-4 hours to converge.

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
