[Supplementary Material]

# A  Experiment Details

## A.1  Hyperparameters

In this section, we list out all the selected hyperparameters in our experiments for reproducibility in Table 1 and Table 2. For selecting the number of layers in the GS and the segmented transformer networks. We did a hyperparameter sweeping of [2, 4, 6, 8] network layers for both GS and the segmented transformer network and find the optimal hyperparameters as presented in below table.

Table 1: Hyperparameters for Policy Network. $gs\_layers$: GraphSAGE layers, $gs\_knn$: GraphSAGE maximum neighbors, $trf\_d\_model$: Dimension of the segmented transformer model, $trf\_n\_head$: Number of attention heads, $trf\_layers$: Number of transformer layers, $trf\_d\_heads$: Dimension of each attention head, $trf\_d\_inner$: Dimension of inner hidden size in positionwise feed-forward.

| Parameters | Value | Parameters | Value |
|---|---|---|---|
| $gs\_layers$ | 4 | $gs\_dim$ | 128 |
| $gs\_knn$ | 5 | $trf\_layers$ | 4 |
| $trf\_d\_model$ | 128 | $trf\_n\_head$ | 3 |
| $trf\_d\_head$ | 15 | $trf\_d\_inner$ | 512 |

Table 2: Hyperparameters for PPO.

| Parameters | Value | Parameters | Value |
|---|---|---|---|
| $learing\ rate$ | 0.5 | $num\ of\ rollouts$ | 800 |
| $minibatches$ | 40 | $epochs$ | 20 |
| $epsilon$ | 0.2 | $entropy$ | 0.5 |
| $optimizer$ | Adam | | |

## A.2  Input Graphs

We used important workloads that are widely used in real applications or are incorporated in industry standard benchmarks like MLPerf. These include ResNet, InceptionNet, WaveNet, Transformer-XL, AmoebaNet, NMT, and RNNLM. In this section, we give a detailed explanation on the selected models and hyperparameters.

### A.2.1  Inception-V3

Inception-V3 [6] is a multi-branch convolutional network used for a variery of computer vision tasks, including classification, recognition, or generation. The network consists of blocks made of multiple branches of concolutional and pooling operations. Within a block, the branches of ops can be executed in parallel. However, the model is mostly sequential as the outputs of each block are concatenated together to form the input to the next block. We use a batch size of 64. The TensorFlow graph of this model contains 24,713 operations.

### A.2.2  AmoebaNet

AmoebaNet [5] is an automatically designed neural network that yields SOTA performance on ImageNet. Similar to Inception-V3, it contains Inception-like blocks called cells, which receives a direct input from the previous cell and a skip input from the cell before it. The network is made of redundant cells stacked together, therefore is more modular than Inception-V3. We use a batch size of 64. The TensorFlow graphs contains 9,430 operations.

### A.2.3  RNNLM

Recurrent Neural Network Language Model [10, 4] is made of many LSTM cells organized in a grid structure. The processing of each LSTM cell only depends on the results of 2 other cells (from the previous layer, and from the previous time step), which make the concurrent execution of many LSTM cells possible given enough hardware resources. We use batch size 64 and a hidden size of

2048. The corresponding TensorFlow graph contains 9,021 operations for a 2-layer model. The number of ops grow roughly proportional with the number of layers.

### A.2.4 GNMT

Neural Machine Translation with attention mechanism [1, 9] has an architecture similar to that of RNNLM, but its many hidden states make it far more computationally expensive than RNNLM. To reduce the training time, prior work [9] propose placing each LSTM layer, as well as the attention and the softmax layer, on a separate device. This strategy demonstrates early success in human placement, we show that GO can find significantly better placements. We use batch size 64. The original 2-layer encoder-decoder consisting of 28,044 operations. An extended 4-layer version consisting of 46,600 operations, An even larger 8-layer version consisting of 83,712 operations.

### A.2.5 Transformer-XL

Transformer-XL [2] is an modified version of Transformer [8] that supports segement-level recurrence and a novel positional encoding scheme. This innovation enables learning dependency that is 80% longer than RNNs, and 450% longer than vanilla Transformers. We use a transformer-XL with batch size of 64, sequence length of 256, segment length of 64, model hidden dimension of 500 and feed forward hidden dimension of 1000, 10 heads, and head dimension of 50. The 2-layer Transformer-XL contains 2,618 operations. The number of ops grow roughly proportional with the number of layers.

### A.2.6 WaveNet

WaveNet [7] is a generative model for speech synthesis. The model is fully probabilistic and autoregressive, with the predictive ditribution for each audio sample conditioned on all previous ones. Architecturally, WaveNet uses causal convolutions with dilations, to obtain a large receptive field. We use a WaveNet model with batch size 64 and a receptive field size of 2048 (9-layers per stack). An 5-stack WaveNet contains 4,374 operations and a 10-stack WaveNet contains 8,516 operations.

### A.3 Performance Model

An analytical performance model [3] based on the roofline model is used in this work. Specifically, the execution time of a kernel is estimated using analytically modeled flops and bytes of memory accessed, together with GPU's peak achievable FLOPS and memory bandwidth. The kernel launching overhead is not considered. For data transfer between devices, the transfer time is estimated using the tensor size and bandwidth. A virtual scheduler is developed to handle the dependency among different ops in the graph within and across devices with several available scheduling policies including the priority based one used in this work. The accuracy of the performance model has been validated against true runtime measurements (on actual hardware) on several industry standard models, including MLP, CNN, RNN, LSTM, Transformer, BERT, etc.

## B  Ops Scheduling

### B.1  Default Scheduling is Sub-optimal When Coupled with Device Placement

Many of the large machine learning models require partitioning of the dataflow graph for the model over a set of heterogeneous devices (like CPUs and GPUs). This partitioning is necessary to overcome the memory limitations of a single device while also reducing the step time (i.e., execution time of a single training step) by exploiting the inherent parallelism in the model architecture. However, the best partitioning of the graph over a set of devices is a complex non-differentiable function of the graph structure and computation capabilities of the devices. Furthermore, even for a seemingly good partitioning and placement of the graph, poor choices in scheduling operations can often lead to near-sequential execution of the partitioned blocks and therefore poor runtimes and low device utilization. For instance, consider the timelines for a single step of Transformer XL model with identical partitioning and placement shown in Figure 1. While the first timeline schedules operations first-in-first-out (FIFO), the second timeline uses a human expert provided schedule. By carefully choosing the ordering in which operations are scheduled, we can reduce step time (as illustrated

Figure 1: Effect of human expert scheduling on timelines for Transformer-XL workload. Step time is reduced by 25.5% from (a) to (b) as shown by the shorter timeline (compressed x-axis). Higher device utilization is observed in (b), with less idle time (as represented by white space). One layer is placed on each device (each row), with colors denoting segments. Each successive device can start on a segment only after it is completed on the previous device.

through a shorter timeline) as well as increase device utilization (as illustrated through less white space in the timelines).

## B.2 Ops Scheduling for Individual Graphs

**Baseline – Heuristics.** We use *First-In-First Out (FIFO)*, which is the default scheduling policy in TensorFlow, and a static *Fanout heuristic*, which assigns priorities based on the number of dependent operations as baselines. Fanout heuristic aims to be a simplification of heuristics such as critical-path-first, and aims to capture the sentiment of prioritizing operations that other ops depend on. In our experiments, Fanout heuristic utilizes the priority scheduler implementation while FIFO numbers are from the default scheduler.

**Baseline – Human expert optimization.** To show the potential for scheduling, we manually added artificial control dependencies to our workloads. This required delving into timelines, diagnosing scheduling issues, and identifying which of the 49K nodes to add dependencies between. After adding 21 dependencies, we reduced the step time of an 8-layer Transformer-XL model by 25.5% as shown in Figure 1. With FIFO scheduling, we note that operations from different segments are interleaved on each device, which we infer to be the cause of unnecessary waiting on subsequent devices.

**Setup.** We evaluate our approach on the workloads RNNLM [11, 4] and Transformer-XL [2]. These workloads are used for language modeling and represent complex state-of-the-art models with grid-like dependencies that offer opportunities for model parallelism. We use a manual placement of assigning one layer to each GPU, and set the number of layers and thus dictating the size of the model to be equal to the number of GPUs. We use a batch size of 64 for training, and a GPU cluster topology consisting of a single machine with 8 Nvidia Tesla P100s, each with 16 GB of memory and NVLink interconnect.

We ran simulated annealing twice with different random seeds but same starting point, with 5000 iterations each. We trained the RL model with a replay buffer and a rollout of 400 actions per iteration. To speed up the search, we use tf-sim[1] to estimate the step time. It considers operations' running time characteristics and data transfer times on target hardware.

Table 3: Effect of scheduling priorities on reductions in step times (relative to default FIFO policy).

| Workload | Layers | FH | SA | RL |
|---|---|---|---|---|
| RNNLM | 2 | -0.84% | 2.96% | **8.77%** |
| RNNLM | 4 | -0.74% | **12.01%** | 7.85% |
| RNNLM | 8 | -0.59% | **14.65%** | 4.30% |
| Transformer-XL | 2 | -4.86% | 6.30% | **14.46%** |
| Transformer-XL | 4 | -3.17% | 9.27% | **26.98%** |
| Transformer-XL | 8 | -0.05% | **19.27%** | 10.67% |

Figure 2: Effect of scheduling priorities on reductions in step times.

**SA and RL priority scheduling reduce step time.** Table 3 and Figure 2 show step time reductions from our simulated runs, where the step time is the time needed to train one minibatch. We observe that simulated annealing (SA) performs strongly, achieving the most reduction in step time for half of the workloads. SA was difficult to tune to get good results, especially in designing the action space and finding a set of hyperparameters that could work well for all workloads. Given that we did not do such tuning for the RL model, its performance should be considered positively given that it was a more automated solution and still outperformed SA for half the workloads in addition to achieving up to 26.98% over FIFO, which was greater than that of SA's 19.27% over FIFO.

From examining timelines such as those in Figure 1, we expected the potential amount of scheduling badness (and thus maximum possible speedup from scheduling) to grow near linearly with the number of devices. This is consistent with the trend of the maximum achieved speedup scaling with the number of GPUs. We observed that the Fanout heuristic performs worse than FIFO scheduling, underlining the difficulty of designing a heuristic that works well across all real workloads.

## C  Ablation Study

### C.1  Model Architecture

As we discussed model architecture alternatives and explained how we designed the network architecture in Section 4.2 in the main paper, we provide more empirical results compared to a decision network based on MLPs and a decision network based on Graph Attention Networks (GATs). The MLPs is combined with GraphSAGE to provide generalization while the GATs supports generalization naturally. We don't compare with an LSTM or a vanilla Transformer model as they cannot scale to problem size interesting enough. According to Figure 3, for many bigger models (e.g. AmoebaNet, GNMT, 8-layer RNNLM and Transformer-XL), GATs failed to generate any valid graph optimizations. For the workloads they can generate valid optimizations, GATs is on average 8% worse and MLP is on average 11% worse in the optimized graph run time, compared to GO-one. The results also shows that the global attention brings in about 11% additional performance, compared to non-attention based decision network.

### C.2  Fine-tuning Results

We also evaluate a fine-tuning strategy by pre-training the graph embedding and placement network and fine-tuning the network on the down stream tasks. The difference here compared to the main

Figure 3: Learnt Policy Comparison with Alternative Decision Networks.

Figure 4: Normalized run time (step time for the generated placement) and normalized training time (search time) for fine-tuning. Time is normalized to GO-one.

paper Section 5.1 "Generalization Across Graphs" is that we also include the target graphs in the pre-training dataset. When GO-batch is used as a pre-training strategy, the graph embedding and placement network assimilate meaningful graph representations and placement policies from a wide set of graphs, thus can be used as a strong baseline network for fine-tuning on downstream tasks. We compare the generated placement run time and the placement search time, normalized to GO-one. We find that fine-tuning further reduces the the placed graph run time by an average of 5% and placement search time by an average of 86%, compared to GO-one.

## Footnotes

[1]TensorFlow Performance Simulator. Publication in progress.