[Reviews · NeurIPS 2020]

Review 1

Summary and Contributions: This paper presents GO, a reinforcement learning method for optimizing execution of computation graphs. GO can learn to solve various optimization tasks that involve making decisions about each node in a compution graph, such as where to place the node, in what order to schedule it, and what other nodes to fuse with it. GO's neural network architecture combines a graph neural network that learns embeddings for nodes, with a Transformer-based policy network that uses recurrent attention layers to output per-node decisions. This neural network architecture enables GO to learn a policy that generalizes across computation graphs, such that it can be used to make decisions for unseen computation graphs without further training (zero-shot) or with minimial per-graph fine-tuning. The authors present results for device placement of several neural networks, outperforming prior methods on real hardware, and also show results for joint placement, scheduling, and operator fusion using a analytical performance model.

Strengths: + Optimizing ML compilers is an important challenge and the paper describes a promising approach backed by solid empirical results. + The proposed neural network architecture is sound and seems like a good fit for the problem. + The generality of the approach and the way it can be used to optimize across multiple tasks jointly is interesting.

Weaknesses: - The design is synthesized from existing components (GraphSage, Transformer, etc.) and has limited novelty from an ML perspective. - The paper could have done a better job at teasing apart the impact of some of the decisions like the use of recurrent attention layers in the policy network and parameter superposition to improve generalization.

Correctness: No major problems, but the evaluation of scheduling and operator fusion using a simple analytical model is not entirely convincing.

Clarity: The paper is generally well-written and motivates the design well. Just a few suggestions: - The discussion of operator fusion could be made more clear. In particular, I could quite follow how "fusion priorities" are used to make fusion decisions. - Equation 2 seems to imply that each Transformer layer in the policy network will output an action per node, and the actions of subsequent layers take actions of previous layers as input. However, it is unclear from Figure 3 and the description whether each Transformer layer actually outputs actions (I'm guessing each layer only outputs a feature embedding that is used by the next layer and only the final layer computes actions). - In section 4.3, how is the modulation layer different from the first Transformer layer in the original architecture (without modulation layer)? In other words, why can't the first Transformer layer do the same thing that you want the modulation layer to learn? - More generally, it would be helpful to formally describe the full neural network model in equations (perhaps in the appendix). As it stands, I'm not sure it would be possible to reproduce the results.

Relation to Prior Work: The paper discusses and differentiates against related work clearly.

Reproducibility: No

Additional Feedback: Please see above.


Review 2

Summary and Contributions: The paper proposes GO, a graph optimizer to bring significant performance benefits in ML compilers on three tasks: device placement, op fusion, and scheduling. The technique (1) leverages GraphSAGE to embed the information in the dataflow graph, and trains the model using PPO; (2) Transformer-XL to keep track of the node dependencies; (3) recurrent attention policy network that improves scalability while increasing the number of tasks.

Strengths: + The paper focuses on an important topic ML for Systems which has been largely dominated by human intervention and heuristics. The paper may help improve the overall productivity in the systems community! + Different ML techniques have been put together elegantly to achieve good results on multiple tasks, including impressive results compared to state-of-the-art and off-the-shelf framework (TF)

Weaknesses: - It is rather difficult to reproduce the results given no open-source implementation. - he evaluation seems to have used TF-Sim, which does not seem to be publicly available. Therefore, fidelity of the evaluation (at least partially) banks on that infrastructure, which is not possible to verify.

Correctness: * The paper makes a lot of effort to provide empirical results to back up their proposed framework, including the ablation studies provided in the supplementary material.

Clarity: The paper is very well written, and the experimental results are well presented.

Relation to Prior Work: The papers seem to have placed their work well in between various works such as Placeto, HDP, and etc.

Reproducibility: No

Additional Feedback: I very much enjoyed reading the paper as this paper. Authors have elegantly placed together recent ML models to solve important problems in the systems domain. Questions 1. In section 4.2, the authors claim "GNNs lack the ability of tracking global node dependencies in a scalable fashion'. The statement would benefit from more detailed description? 2. How would this be trained in an on-line scenario similar to the one presented in the Decima [1]? [1] Mao, Hongzi, et al. "Learning scheduling algorithms for data processing clusters." Proceedings of the ACM Special Interest Group on Data Communication. 2019. 270-288. ========== Post Rebuttal Comments ========== I thank the authors for their response. In good faith and on the premise of the authors' plans to open-source the project, I am willing to boost the score. I look forward to the revision and the artifact!


Review 3

Summary and Contributions: The paper presents an end-to-end differentiable model for optimizing deep learning graphs. The model predicts device placement, scheduling, and fusion priority. The model is based on a graph embedding followed by attention layers, and is trained with Proximal Policy Optimization. The model trained on each different task is evaluated on a set of popular machine learning models, showing that it results in faster computation graphs and less search time than the current approaches for optimization of these tasks.

Strengths: The paper is very well motivated, fairly well written, and has very compelling results, significantly outperforming the current standard practice (Tensorflow default) and prior work like HDP along all axes. The model is implemented with minimal explicit featurization, meaning that the results do seem to be based on learning relevant properties of the computation graph.

Weaknesses: - The paper has some unjustified claims interspersed throughout the paper (noted below in the "Correctness" section) - The paper uses an analytical performance model without validating that the predictions from the performance model are accurate — this means it's hard to have confidence in the results that use the performance model - The exact details of the model are somewhat hard to understand

Correctness: - Table 1: is there a methodology for these yes/no distinctions? - Line 172: "... from realistic workloads consisting over 10,000 nodes.": is there a source for the claim that these are "realistic" workloads? To validate that this is required, it would be worthwhile to include the node count of each benchmark in Table 2 (moving them up from the appendix). - Run Time Measurement: the paper presents measured end-to-end speedup with device placement, but simulated results for fusion and scheduling. Such performance models are notoriously inaccurate, and can easily have systematic biases that a learned model could overfit. Without details on the accuracy of the performance model, it is hard to take anything useful away from these simulated results. - There are some claims in the introduction that are not evaluated: "GO converges faster using less resources" (line 54). "without introducing ... training overhead" (line 58). - "Compared with SA, which takes 20x longer to search" (line 279). What are the search times for SA? What are the search times for GO on this task?

Clarity: - Section 4 and Figure 3 are fairly hard to parse. Following are some suggestions for how to make this more clear: -- It would help if Figure 3 were annotated to include the subsection in which each component is discussed. -- There are also some components in the diagram that are not explicitly referenced in the text as far as I can tell (e.g., "state embedding"). It would help if each component were given a canonical name, and the text always used that canonical name to refer to that component. -- There is history and justification for the different components intermingled with the specification (e.g., the first paragraphs of 4.2). Given the complexity of the network, it might be better if there is a compact technical specification of the network, followed by (or preceded by, but not interspersed with) justification of the design decisions.

Relation to Prior Work: It is clearly discussed how this work differs from previous contributions.

Reproducibility: No

Additional Feedback: I have marked that the major results are not reproducible because the performance simulator is proprietary, and some details of the model and evaluation are missing (e.g., the exact features that are used in the node embedding on line 152, how exactly fusion is performed based on the prediction of the model, the version of Tensorflow and other details of the system environment in the evaluation). Other miscellaneous feedback: - Line 175: an inline definition of "hierarchical groupings" would help for those unfamiliar with the term. - Table 3: the column with network and compute environment is labeled "Speedup". Is this a typo? - Figure 1: Some notion of syntax for these graphs would help — this seems to rely on some knowledge that isn’t presented (what are green and red, how is it possible to fuse when there are other nodes that use K1’s output) - Line 112: Needs a bit more explanation — what exactly is the relationship between “order” and fusion? Do you always fuse every op that can be fused, but just in the specified priority order? Is it possible to decide against fusion? - Line 161: is this function $f^{(l+1)}$ different each iteration? - Line 118: The syntax of "J(theta) = ..." is a bit confusing, specifically "T ~ pi_theta(G)": what does it mean to sample T \in {D, P, F} from pi_theta(G)? Does this policy say which task to run? ====================== Update after author response: ====================== Thanks to the authors for the response to the reviews. Based on the response, I plan to keep my score as-is at a 7. Although I still wish there was better methodology for some of the claims made in the paper (Table 1), the author have otherwise addressed my concerns about validation of the performance model and clarity of the presentation.


Review 4

Summary and Contributions: The paper proposed an end-to-end graph optimization which make decisions over the graph. The performance significantly improves the default TensorFlow optimization.

Strengths: 1. The method is effective and has great potential to practical deployment.

Weaknesses: 1.The authors proposed to use RL based method, while the motivation is not very clear. 2. It's hard to reproduce the results. Will the code be public avaliable.

Correctness: yes

Clarity: yes

Relation to Prior Work: yes

Reproducibility: No

Additional Feedback:

[Author Response · NeurIPS 2020]

We thank all reviewers for their encouraging and constructive feedback.

**Rev#1**: We thank the reviewer for their suggestion to tease apart the impact of architectural decisions, have a clearer
elaboration of the fusion task, and include full neural network model formulation. We will include additional details
and ablation studies in the revised version to further clarify each of these aspects.

**[Fusion priority]** As mentioned in Sec 3 of the paper, each node is associated with a fusion priority score (ranging
from 1 to $F$, where $F$ is a hyperparameter). The scores determine the order in which nodes are fused.

**[Better figure]** The reviewer is correct that each Transformer layer in Fig.3 only outputs a feature embedding that is
used by the next layer, and only the final layer computes actions. In Eq.2, the actions are produced by the final layer in
Fig.3. However, the temporal dependencies between actions in multiple RL steps are carried out by state embedding
such that the state of the previous step is used as an input to the network in the next step. Fig.3 is an illustration of
a single task, while Fig.4 demonstrates the extension of the same framework for multiple tasks. We add a few more
Transformer layers with residual connections as task heads. The yellow part in Fig.4 is the same as Fig.3.

**[Why not use the first layer of the TRF to replace the modulation layer?]** Having a separate modulation layer
provides additional parameterization to the network and hence better orchestration of the training of the parameters. It
empirically results in better generalization.

**Rev#2**: We thank the reviewer's insightful feedback on providing more details and potential open sourcing for
reproducibility. We will add more detailed explanations about GNN's limitations on tracking global node dependencies.

**[Reproducibility]** We made an extensive effort to document the details of our network architecture, the input graphs,
and other hyperparameters in Sec. 4 and Supp.A. We plan to incorporate further details to enhance reproducibility and
open source the GO framework along with the performance model. We would also like to emphasize that GO, as a
general algorithm, can be applied to problems at different layers of the compilation stack, even on different platforms.
We have successfully applied GO to two other graph optimization problems at different stages in the compiler stack –
tensor layout optimization and operator fusion in XLA (as opposed to at the TF level discussed in this paper).

**[Can GO be trained online?]** Yes, GO can be trained in an on-line scenario similar to Decima. Specifically, GO-one
is quite similar to an online training method where state transitions and rewards can be collected by interacting with the
compiler. However, as pointed out, an online RL can make poor decisions in early stages of training and the quality
of training is subjective to input data distribution. This is particular challenging for a compilation problem, where a
program needs to be compiled within a short time. One possible approach is to pre-train the model like GO-finetune,
then apply online training to continuously improve the policy.

**Rev#3**: We appreciate the reviewer's detailed feedback and will address the questions in the final version.

**[Reproducibility and validation]** Please see our response to Rev#2. The accuracy of the performance model has been
validated against true runtime measurements (on actual hardware) on several industry standard models, including MLP,
CNN, RNN, LSTM, Transformer, BERT, etc. We will provide more validation results in the final paper.

**[Table1: is there a methodology for these yes/no distinctions?]** We empirically observed these distinctions. For
example, a vanilla Transformer network will be OOM for problems larger than 10k nodes, and a vanilla LSTM cannot
generate placement decisions for input sequence longer than a few hundreds. Ablation study in Supp.C.1 addresses
some of the distinctions.

**[Source for the claim the workloads are realistic]** We used important workloads that are widely used in real
applications or are incorporated in industry standard benchmarks like MLPerf. These include ResNet, InceptionNet,
WaveNet, Transformer-XL, AmoebaNet, NMT, and RNNLM.

**[SA convergence time]** SA takes around 24 hours to converge in our evaluated tasks. For a smaller graph, GO takes
less than an hour to converge. For a large graph over 10k nodes, GO can take 1-4 hours to converge.

**[Better figure]** We acknowledge that Figure 3 can be improved, and we will incorporate the suggestions from the
reviewer to the final version.

**[More details of the performance model]** Exact features used in node embedding (as presented in Fig.3 dashed box):
tensor shape, op type, and adjacency matrix. Fusion is optimized using a priority-based node traversal. Please also see
reponse to Rev#1 [Fusion priority].

**[Is this function $f^{(l+1)}$ different each iteration?]** Yes. $\ell$ indexes the layer of the graph neural net and different layers
of the graph neural net have different $f$.

**Rev#4**: As mentioned in response to Rev#2, we plan to open source the GO framework along with the performance
model, and will include details about this in the final version.

[Meta-Review · NeurIPS 2020]

This paper describes a well-motivated and elegant method for optimizing execution of neural network computation graphs using reinforcement learning. The paper is well-written, however there is no open source implementation, hence results may be difficult to reproduce.